# Breaking the limitation of mode building time in an optoelectronic oscillator

Tengfei Hao[1,2], Qizhuang Cen[3], Yitang Dai[3], Jian Tang[1,2], Wei Li[1,2], Jianping Yao [4], Ninghua Zhu[1,2] & Ming Li [1,2]

An optoelectronic oscillator (OEO) is a microwave photonic system with a positive feedback loop used to create microwave oscillation with ultra-low phase noise thanks to the employment of a high-quality-factor energy storage element, such as a fiber delay line. For many applications, a frequency-tunable microwave signal or waveform, such as a linearly chirped microwave waveform (LCMW), is also needed. Due to the long characteristic time constant required for building up stable oscillation at an oscillation mode, it is impossible to generate an LCMW with a large chirp rate using a conventional frequency-tunable OEO. In this study, we propose and demonstrate a new scheme to generate a large chirp-rate LCMW based on Fourier domain mode locking technique to break the limitation of mode building time in an OEO. An LCMW with a high chirp rate of 0.34 GHz/μs and a large time-bandwidth product of 166,650 is demonstrated.

[1] State Key Laboratory on Integrated Optoelectronics, Institute of Semiconductors, Chinese Academy of Sciences, Beijing 100083, China. [2] School of Electronic, Electrical and Communication Engineering, University of Chinese Academy of Sciences, Beijing 100049, China. [3] State Key Laboratory of Information Photonics and Optical Communications, Beijing University of Posts and Telecommunications, Beijing 100876, China. [4] Microwave Photonics Research Laboratory, University of Ottawa, Ottawa, ON K1N 6N5, Canada. These authors contributed equally: Tengfei Hao, Qizhuang Cen, Yitang Dai. Correspondence and requests for materials should be addressed to J.Y. (email: jpyao@uottawa.ca) or to N.Z. (email: nhzhu@semi.ac.cn) or to M.L. (email: ml@semi.ac.cn)

A linearly chirped microwave waveform (LCMW) with a large time bandwidth product (TBWP) is widely used in modern radar and wireless communication systems[1–4]. For instance, in a radar system[1, 2], an LCMW with a large TBWP is used to increase the range resolution by performing pulse compression or matched filtering at the radar receiver, which overcomes the tradeoff between the range resolution and the detection distance existing in a traditional radar system. In addition, the use of an LCMW permits a pulse to have much longer time duration, which would provide a much higher average power, making the detection distance increased. Furthermore, the signal-to-noise ratio (SNR) can be significantly increased by matched filtering. In a frequency-hopping communication system, an LCMW with a fast tuning speed and large bandwidth is also desired to enhance the anti-reconnaissance and anti-jamming capability[3, 4]. Photonics has been extensively investigated to generate high-frequency and wide-bandwidth microwave waveforms[5, 6] with low phase noise. Those techniques are implemented based on photonic techniques, such as opto-electronic oscillation[7, 8], optical frequency division[9, 10], Kerr frequency comb oscillation[11], wavelength or space to time mapping[12–14], and temporal pulse shaping[15–17]. Although those photonic-assisted techniques can generate microwave waveforms at a high frequency, the bandwidth and the temporal width are usually limited. In addition, the phase noise performance is deteriorated if the bandwidth is increased. Recently, photonic generation of a microwave waveform with a TBWP as large as 120,000 has been demonstrated based on the period-one oscillation in an optically injected semiconductor laser[18–20]. An optoelectronic feedback loop is incorporated to reduce the noise of the generated microwave waveform[19, 20]. However, the generation of period-one oscillation is much strictly restricted by the injection conditions such as frequency detuning and injection power ratio between the master and slave lasers[21]. Moreover, the frequency of the generated microwave waveform is extremely sensitive to the polarization state of the master light. Therefore, it is important to find solutions to generate a high-frequency microwave waveform with ultra large TBWP and low phase noise[22, 23].

Among the photonic-assisted techniques, microwave waveform generation based on an optoelectronic oscillator (OEO) is a simple and cost-effective solution to generate a low phase noise microwave signal thanks to the use of a long and a low-cost optical delay line[7, 8, 24–29]. A long optical delay line in an OEO can make the OEO loop have a high-quality-factor (Q-factor), which would ensure low phase noise. However, a long OEO loop would make the oscillation to have a long mode building time. To generate a frequency-tunable microwave signal, the frequency scanning speed must be low, limited by this long mode building time. In addition, the phase relationship between adjacent oscillation modes is not fixed, which would deteriorate the phase noise performance of the generated frequency-tunable microwave signal.

To generate a fast frequency tunable microwave signal, in this paper, we propose a new scheme based on Fourier domain mode locking (FDML)[30] to break the limitation of mode building time. A LCMW with a large chirp rate and long temporal duration leading to a large TBWP is generated. The fundamental concept is to stimulate thousands of longitudinal modes simultaneously with fixed phase relationship in the Fourier domain, or equivalently in the time domain, to force a periodically repeated and stable chirped oscillation directly in an opto-electronic cavity. Thus, the output frequency can be tuned at an ultra-fast speed. The proposed scheme is analyzed theoretically and verified by an experiment. A LCMW with a chirp rate as large as 0.34 GHz/μs and a TBWP as high as 166,650 is experimentally generated.

## Results

**Principle.** Figure 1 shows the operations of a conventional OEO and an OEO based on FDML. As can be seen from Fig. 1a, a conventional OEO consists of a laser source, a modulator, a photodetector (PD), a broadband gain device, and a narrowband microwave filter or a microwave photonic filter (MPF) in the cavity[7]. The frequency of the generated microwave signal can be tuned by tuning the center frequency of the microwave filter. The maximum achievable frequency tuning rate is limited by the characteristic time constant or mode building time for building up oscillation in a new oscillation mode in the cavity. This non-stationary operation corresponds to temporally varying the distribution of energy from one longitudinal mode to another in the cavity during frequency tuning.

Figure 1b shows an OEO for fast frequency tuning based on FDML. The operation is similar to a wavelength-swept laser source[30], where the wavelength tuning is implemented based on FDML in the optical domain. In the proposed OEO, the frequency tuning is implemented based on FDML in the electrical domain. In the proposed system, a fast tuning MPF is incorporated in the OEO loop for frequency tuning, which is done by using a periodic driving signal to control the MPF to make the frequency tuning period or its multiple be synchronized with the round-trip time of the OEO loop. This process produces a quasi-stationary operation[30] where each instantaneous

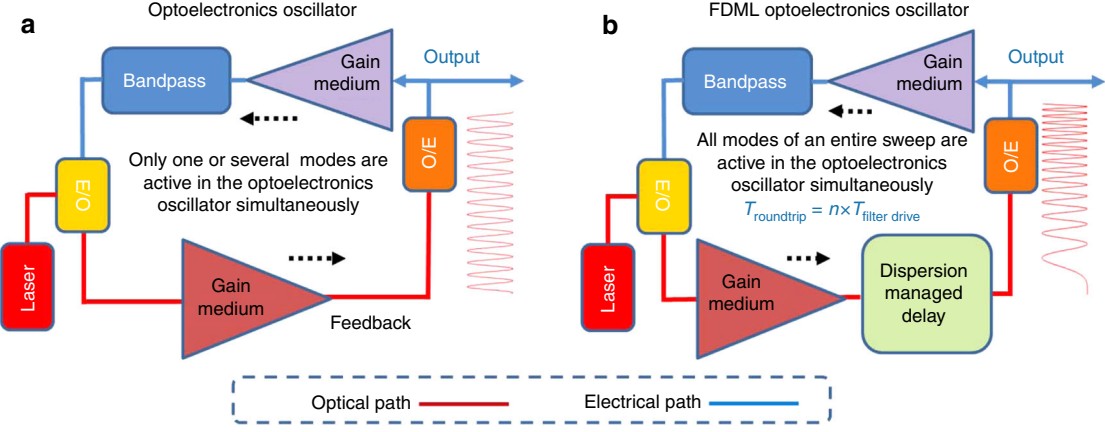

**Fig. 1** Schematic to show the operations of a conventional OEO and an OEO based on FDML. **a** A conventional single-frequency OEO, only one mode is active in the cavity. **b** An OEO based on FDML for generation of a microwave signal with fast frequency tuning, all modes are active in the cavity. E/O electrical to optical conversion; O/E optical to electrical conversion

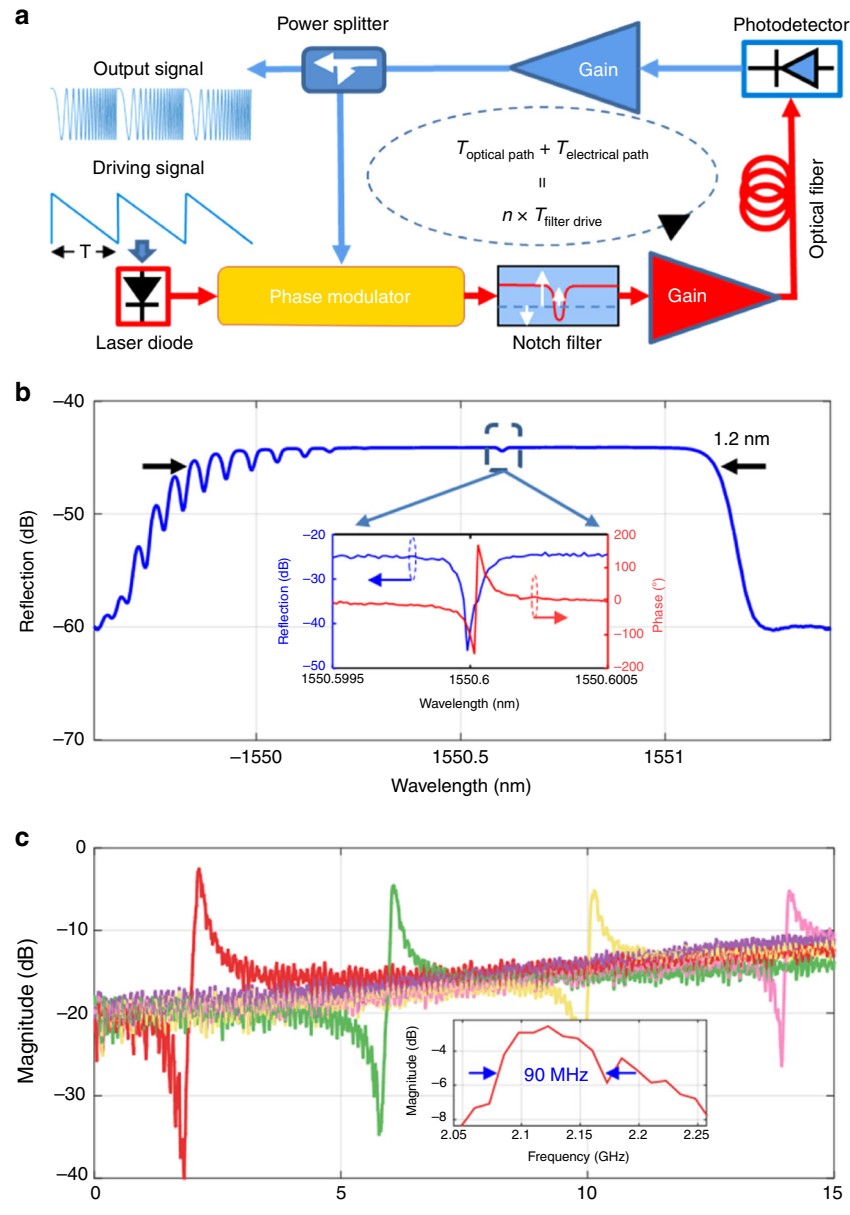

**Fig. 2** Experimental setup and microwave photonic band-pass filtering principle. **a** Experimental setup of the FDML OEO. Light from a laser diode is phase modulated and then is directed into an ultra-narrow optical notch filter. Oscillation frequency is determined by the wavelength difference between the carrier wavelength and notch wavelength. A frequency-selected microwave is generated at the output of a photodetector, and amplified in a gain medium. Part of the amplified microwave signal is fed back to the phase modulator to close the OEO oscillation loop. **b** Reflection spectrum of the PS-FBG. The inset shows the phase and magnitude responses around the notch. **c** Frequency response of the microwave photonics band-pass filter with the central frequency tuned from about 2 GHz to about 14 GHz with a tuning step of about 4 GHz. The inset shows the zoom-in view of the frequency response when the center frequency is tuned at about 2.1 GHz

frequency component propagating through the cavity will return to the MPF at exactly the same time as the passband of the MPF is tuned at the same position, thus all longitudinal modes are active simultaneously for the entire sweep. On the other hand, in the frequency domain, the phase relationship among the longitudinal modes is maintained fixed. Thus, the process is to produce a fast frequency-scanning microwave signal, which is in fact a highly frequency chirped microwave waveform.

Figure 2a shows the experimental setup. The cavity contains three parts, a radio frequency (RF) and an optical amplifier, a long time delay line, and a fast frequency-tunable MPF. The long time delay line is a dispersion-free 4.5 km optical fiber consisting of a single-mode fiber and a dispersion compensating fiber with

opposite dispersion of the same magnitude. The optical fiber has a residual dispersion of −0.848 ps/nm and total insertion loss of 3.8 dB. To implement the MPF, a light wave from a tunable laser source (TLS) with 13 dBm output power is sent to a 20-GHz phase modulator (PM). A phase-shifted fiber Bragg grating (PS-FBG) with an ultra-narrow notch is used to remove one of the two sidebands. The reflection spectrum of the PS-FBG is given in Fig. 2b, where the inset shows the magnitude and phase responses of the PS-FBG around the notch. The notch is located at around 1550.6 nm with a single phase cycle as shown in the phase response. The output light wave from the PS-FBG is amplified by an erbium-doped fiber amplifier (EDFA, JDS Uniphase) with a small signal gain of 37 dB and a noise figure of 3.3 dB. A PD with

a 3-dB bandwidth of 15 GHz and a conversion gain of 300 V/W is used to convert the optical signal to an electrical signal. The overall operation is equivalent to a high-Q MPF, implemented based on phase modulation, and phase-modulation to intensity-modulation (PM-IM) conversion[31, 32]. The frequency tuning of the MPF is achieved by sweeping the wavelength of the TLS which is driven by a sawtooth current. An electrical low-noise amplifier (LNA) with a gain of 22 dB and a noise figure of 6 dB is used to boost the power of the electrical signal. The amplified electrical signal is fed back to the PM to form a closed loop.

According to the theory about PM–IM conversion given in Supplementary Note 1, the signal at the output of the tunable MPF is given by

$$V_{\mathrm{OUT}}^{\Omega}(t) = T\left(\left|V_{\mathrm{IN}}^{\Omega}\right|\right)\left[\left(V_{\mathrm{IN}}^{\Omega}(t)e^{i\varphi_{\mathrm{OC}}(t)}\right) * s_{21}^{\mathrm{openloop}}(t)\right]e^{-i\varphi_{\mathrm{OC}}(t)} \\ + n_A(t) \tag{1}$$

where $V_{\mathrm{IN}}^{\Omega}(t)e^{-i\Omega t}$ and $V_{\mathrm{OUT}}^{\Omega}(t)e^{-i\Omega t}$ are the input and output microwave signals, respectively, and $*$ is the convolution operation. $T\left(\left|V_{\mathrm{IN}}^{\Omega}\right|\right)$ is the saturation factor of PM–IM conversion and $T\left(\left|V_{\mathrm{IN}}^{\Omega}\right|\right) = 2J_0\left(\pi\left|V_{\mathrm{IN}}^{\Omega}\right|/V_{\pi}\right)J_1\left(\pi\left|V_{\mathrm{IN}}^{\Omega}\right|/V_{\pi}\right)/\left(\pi\left|V_{\mathrm{IN}}^{\Omega}\right|/V_{\pi}\right)$. $T\left(\left|V_{\mathrm{IN}}^{\Omega}\right|\right) = 1$ for small-signal approximation. $\varphi_{\mathrm{OC}}(t)$ is the phase variation of the continuous wave (CW) light source. $n_A(t)$ is an additive noise induced by the optical and RF amplifiers as well as the intensity noise of the light source. Assuming a CW light with a constant phase (i.e. $\varphi_{\mathrm{OC}}(t) = $ constant) one can see that $s_{21}^{\mathrm{open\ loop}}(t)$ is the inverse Fourier transform of the frequency response of the MPF when it is static. Equation (1) indicates that a time-varying, frequency-scanning filter could be expressed mathematically by a "convolutional filter" where the input and output of a time-invariant filter are frequency down-converted and then up-converted by the same local oscillation. The central frequency of the MPF can be adjusted by tuning the wavelength spacing between the wavelength of the TLS and the notch wavelength of the PS-FBG. As shown in Fig. 2c, a tunable MPF with a 3-dB bandwidth of 90 MHz is realized which is measured by a vector network analyzer. A frequency-selected microwave signal which is generated at the PD is then amplified and feedback to the PM to close the oscillation loop.

By driving the TLS with a saw-tooth current, the wavelength of the light wave could be linearly and fast tuned[33], to change the oscillation microwave frequency. The frequency-scanned oscillation, $V_{\mathrm{FDML}}^{\Omega}(t)e^{-i\Omega t}$, should satisfy

$$V_{\mathrm{FDML}}^{\Omega}(t-\tau) = T\left(\left|V_{\mathrm{FDML}}^{\Omega}\right|\right)\left[\left(V_{\mathrm{FDML}}^{\Omega}(t)e^{i\varphi_{\mathrm{OC}}(t)}\right) * s_{21}^{\mathrm{open\ loop}}(t)\right]e^{-i\varphi_{\mathrm{OC}}(t)} + n_A(t) \tag{2}$$

where $\tau$ is the time delay of the loop. Equation (2) shows that a stable solution would repeat itself after the loop round-trip delay. In an FDML OEO, the light from the TLS is frequency-tuned periodically, so that it is reasonable to describe the light wave as $\varphi_{\mathrm{OC}}(t) = \varphi_{\mathrm{OC}}^{\tau}(t) + \theta_{\mathrm{OC}}(t)$ where $\varphi_{\mathrm{OC}}^{\tau}(t)$ is the target periodic phase variation with $\varphi_{\mathrm{OC}}^{\tau}(t) = \varphi_{\mathrm{OC}}^{\tau}(t-\tau)$ and $\theta_{\mathrm{OC}}(t)$ is the phase noise. Without any noise, stable oscillation satisfies

$$V_{\mathrm{FDML}}^{\Omega}(t-\tau) = T\left(\left|V_{\mathrm{FDML}}^{\Omega}\right|\right)\left[\left(V_{\mathrm{FDML}}^{\Omega}(t)e^{i\varphi_{\mathrm{OC}}^{\tau}(t)}\right) * s_{21}^{\mathrm{open\ loop}}(t)\right]e^{-i\varphi_{\mathrm{OC}}^{\tau}(t)} \tag{3}$$

according to Eq. (2). Equation (3) indicates that a stable oscillation passes through devices in the opto-electronic cavity and then recovers itself, while obtains the total cavity delay $\tau$. Obviously $V_{\mathrm{FDML}}^{\Omega}(t) \propto e^{-i\varphi_{\mathrm{OC}}^{\tau}(t)}$ is a solution. Such solution reverses any frequency variation of the TLS, and the results in a constant sideband at the notch center [i.e. $V_{\mathrm{FDML}}^{\Omega}(t)e^{i\varphi_{\mathrm{OC}}^{\tau}(t)}$ in Eq.

(3)]. We then conclude that when the roundtrip time (i.e., $T_{\mathrm{optical\ path}} + T_{\mathrm{electrical\ path}}$) equals to $n$ times of the period of the TLS driving signal, the condition for achieving FDML optoelectronics oscillation is satisfied. Thus, a continuous frequency-scanning microwave signal will be generated, and the repetition rate of the generated microwave waveform follows that of the laser driving source, i.e., periodic-chirped light wave generates RF signal with exactly-the-same period and chirp rate RF signal. The periodic microwave waveform results in discrete and equally-spaced frequency modes in the Fourier domain. All the modes are stably oscillating together, and are mode-locked with fixed phase relationship among them. Such an operation is different from a conventional single-mode OEO. Thus, the mode building time is not required and thus fast frequency scanning is possible. Temporal traces of the oscillation process of FDML OEO can be found in Supplementary Note 2.

**Experiment**. To investigate the performance of the proposed FDML OEO, an experiment is performed in which an X-band [i.e. 8–12 GHz] frequency scanning microwave waveform is generated. Figure 3a shows the measured spectrum of the generated microwave waveform. It has a frequency range from 8 to 12 GHz. Figure 3b displays the spectrum at around 10 GHz with a span of 200 kHz. As can be seen the frequency spacing between two adjacent modes is 45 kHz which is determined by to the loop length of the FDML OEO. Figure 3c shows the generated microwave waveform in the time domain, which is measured by a high-speed digital phosphor oscilloscope (Tektronix) with a sampling rate of 100 GS/s. The waveform has a temporal period of 22.22 μs. Considering the bandwidth of 4 GHz, its TBWP is 88,880. The inset in Fig. 3c shows a real-time capture of a section of the generated waveform. The overlay of temporal traces of the X-band frequency scanning microwave waveform can be found in Supplementary Note 3. Figure 3d shows the instantaneous frequency of the generated waveform, which is obtained by calculating the short-time Fourier transform (STFT) of the generated microwave waveform. As can be seen, the waveform is periodic and within one period (22.22 μs) the instantaneous frequency is linearly increasing. The chirp rate is calculated to be 0.18 GHz/μs which is much slower than that reported in other literatures[19, 20]. It is noted that the chirp rate is actually tunable by changing the speed of the linear saw tooth driving signal and the fiber length of the OEO. The product between the tuning speed and TBWP equals to the square of the scanning bandwidth. For a fixed scanning bandwidth, there is a tradeoff between the tuning speed and the TBWP. A large TBWP has been chosen in our experiment since a large TBWP is always required in radar systems.

A slight nonlinearity of the frequency distribution of the generated chirped waveform as shown in Fig. 3d is observed, which has mainly resulted from the nonlinear response of the laser diode to the linear saw-tooth driving signal. The linearity of the chirped waveform can be improved using a feedback control loop[34]. In modern radar systems, pulse compression is generally used to increase the range resolution. This is normally realized using a matched filter with an impulse-response corresponding to the complex conjugate of the time reversal of the received signal. The output of the matched filter is identical to the autocorrelation of the chirped microwave waveform. In order to see the pulse compression capability, we calculated the autocorrelation of the generated LCMW, and the result is shown in Fig. 3e. The width of the compressed pulse is around 0.275 ns, corresponding to a pulse compression ratio of 80,800.

To investigate the reconfigurability of the FDML OEO, experiments are performed in which the saw-tooth driving current is changed by either changing the slope or the direct current (DC)

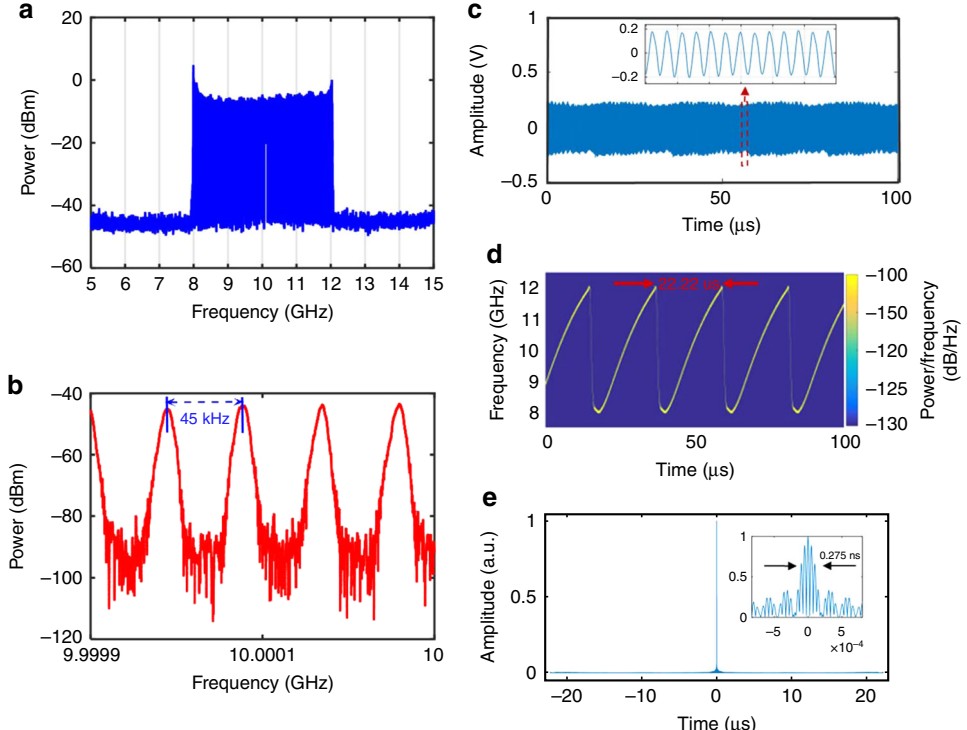

**Fig. 3** Experimental results. **a** Spectrum of a generated X-band frequency-scanning microwave waveform with a span of 10 GHz. **b** Spectrum with a span of 200 kHz. **c** Temporal waveform of the periodically and continuously chirped microwave waveform, the inset shows a section of the waveform. **d** Real-time frequency distribution. **e** The compressed pulse by autocorrelation (inset: zoom-in display)

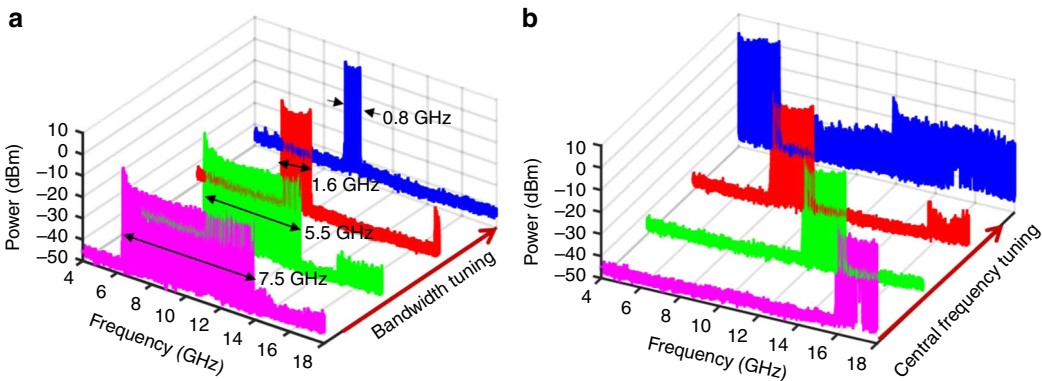

**Fig. 4** Tuning of the generated microwave waveforms. **a** The scanning range is tuned from 0.8 GHz to 7.5 GHz with a central frequency of 10 GHz. **b** The central frequency is tuned from 5 GHz to 17 GHz with a scanning range of 2 GHz

bias, to modify the frequency scanning range or central frequency. Figure 4a shows the change of the frequency scanning range. As can be seen the scanning range is tuned from 0.8 GHz to 7.5 GHz corresponding to a saw-tooth driving current with a magnitude from 1.85 mA to 17.33 mA. The TBWP of the generated chirped microwave waveform is as high as 166,650. By changing the DC bias of the saw-tooth driving current, the center frequency of the generated microwave waveform is changed. Figure 4b shows the tuning of the central frequency of the generated waveform. As can be seen, the central frequency is tuned from 5 GHz to 17 GHz with a frequency interval of 2 GHz by changing the DC current of the driving source. The frequency tuning resolution is around 5 MHz in our experiment, which is limited by the resolution of the DC driving source. The tunable range could be further enhanced by extending the bandwidths of the optical and electrical components such as the PM, PD, the tuning bandwidth of the TLS and the bandwidth of the

PS-FBG. The optical spectra when the OEO operating at different cases can be found in Supplementary Note 4.

The light wave from the TLS contains phase noise which will be translated to the phase noise of the generated microwave waveform. The phase noise performance of the FDML OEO is then theoretically analyzed. When the noise term in Eq. (2) is not zero, especially when the phase noise of the TLS, $\theta_{OC}(t)$, is significant, the oscillation will have noise. The actual FDML oscillation should then be the ideal $e^{-i\varphi^{\tau}_{OC}(t)}$ disturbed by a noisy $V^{\Omega}_{OSC}(t)$

$$V^{\Omega}_{FDML}(t) = V^{\Omega}_{OSC}(t)e^{-i\varphi^{\tau}_{OC}(t)} \qquad (4)$$

According to Eq. (3), $V^{\Omega}_{OSC}(t)$ satisfies

$$V^{\Omega}_{OSC}(t-\tau) = T\left(|V^{\Omega}_{OSC}(t)|\right)\left[\left(V^{\Omega}_{OSC}(t)e^{i\theta_{OC}(t)}\right) * s^{openloop}_{21}(t)\right]e^{-i\theta_{OC}(t)} + n_A(t)e^{i\varphi^{\tau}_{OC}(t)} \qquad (5)$$

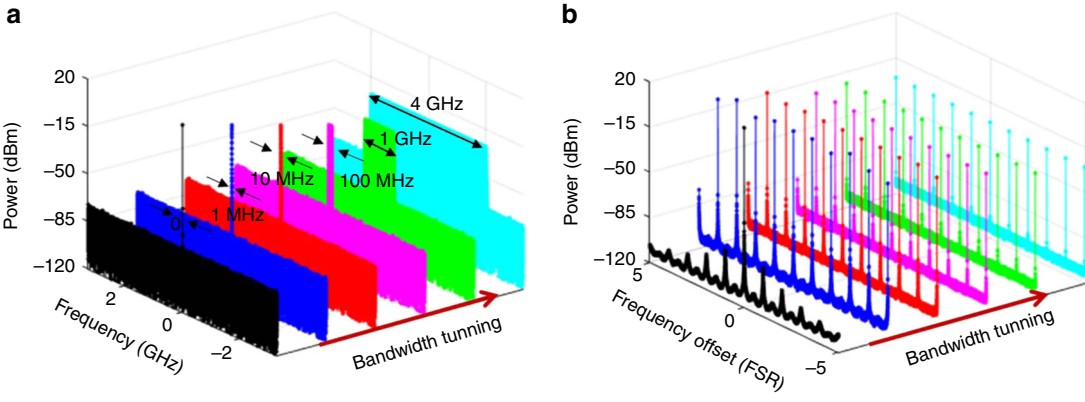

**Fig. 5** Frequency scanning bandwidth vs output power level. **a** Power spectrum when the scanning bandwidth is tuned from 0 to 4 GHz. **b** Zoom-in views of power spectrum when the scanning bandwidth is tuned from 0 to 4 GHz

Comparing Eqs. (2) and (5), one can find that Eq. (5) describes a typical PM–IM-conversion-based single-mode OEO, and the setup is exactly the same as in Fig. 2a except that the frequency of the light source is stable with a phase noise of $\theta_{OC}(t)$. A more detailed theoretical analysis of such a conventional OEO can be found in ref. [24]. Although the light source has significant phase noise, much larger notch bandwidth as well as long fiber delay line can suppress the noise that is transferred to $V_{OSC}^{\Omega}(t)$. Such an expression connects FDML OEO with a conventional OEO, and we can expect that the feature of FDML oscillation is similar in theory with the corresponding conventional signal-mode PM–IM OEO which uses the same optical and electronic components.

We then evaluate the above theory numerically. We assume that the total cavity delay is $\tau_c = 20\,\mu s$, which is also the period of the saw-tooth driving current. The PM has a 7-V half-wave voltage. The bandwidth of the notch filter in our simulation is 90 MHz and has the delay of 3.5 ns. The PD has a responsivity of 0.5 A/W and average optical power injected into on the PD is 9 dBm. The loop loss is about 26 dB. In order to get stable oscillation, a linear electrical amplifier with power gain of 32 dB and a noise figure (NF) of 5 dB is incorporated in the cavity. The additive noise is modeled according to refs. [35, 36], and the noise density at the output of the PD is –153 dBm/Hz. In the simulation, the laser frequency noise is also considered, which has a $f^{-2}$ distribution and is –60 dBc/Hz @ 10 kHz. The OEO model is similar to that reported in ref. [37]. Numerically, an initial random field passes through the noisy microwave-photonic link and the scanning filter, alternately and iteratively, until the field becomes constant after a finite number of cavity traversals. The in-cavity oscillation after the LNA is analyzed.

In the simulation, the laser has different frequency-scanning spans ($B_{scan}$), from zero (corresponding to a conventional OEO) to 4 GHz, and its instantaneous frequency changes linearly from $\omega_0/2\pi - B_{scan}/2$ to $\omega_0/2\pi + B_{scan}/2$ during half scanning period, while back to $\omega_0/2\pi - B_{scan}/2$ after the other half. As the tuning period is synchronized with the cavity delay and net gain in the opto-electronic loop is greater than 1, stable chirp oscillation is achieved.

The generated microwave waveform has the same chirp rate and bandwidth with the frequency-scanning TLS, which is consistent with our theoretical analysis. The average power of stable oscillation is about 19.2 dBm. The power spectra under different $B_{scan}$ are shown in Fig. 5a, their zoom-in views are given in Fig. 5b. One can find that the bandwidth of the output signal follows the $B_{scan}$ and the noise floor remains unchanged. However, as the $B_{scan}$ gets larger, the power of each line becomes weaker, as shown in the zoom-in view of Fig. 5b. As a result, the

single-sideband (SSB) noise floor of each line gets worse. At a frequency-offset close to the carrier, the SSB noise of the FDML OEO is identical for different scanning bandwidths as well as for a single mode OEO (i.e. $B_{scan} = 0$), as shown in Fig. 6, because the phase noise is determined by the Q-factor of the OEO cavity. At a frequency-offset far from the carrier, however, additive noise would be the dominant component in the phase noise. The power of each mode would decrease with the increase in the scanning bandwidth of the FDML OEO. Thus, the phase noise performance would get deteriorated with the increase in the scanning bandwidth. The SSB noise at a frequency-offset far from the carrier would increase approximately by $10\log_{10}(B_{scan}/FSR)$ as compared with that of a single-mode OEO where the FSR is the free spectral range which defined as the frequency interval between two adjacent modes. Although the phase noise performance at a frequency-offset far from the carrier gets deteriorated with the increase in the scanning bandwidth, the SNR is maintained unchanged for different scanning bandwidths. The SNR is defined as the ratio between the total signal power and the total noise power for a fixed observation bandwidth. For a 4-GHz observation bandwidth, the SNR is calculated to be 73 dB for a scanning bandwidth of 4 GHz (from 0 to 4 GHz). The same SNR for different scanning bandwidths indicates that the FDML OEO oscillates in the same way as a conventional single mode OEO, except that the energy is shared by the many oscillation modes. The phase mismatch between the optical and microwave signals in the PM would result in an uneven power distribution of the generated microwave waveform, leading to an increased phase noise at a high frequency offset. In order to reduce the impact of the phase mismatch on the phase noise of the generated microwave signal, we chose a PM with a sufficiently wide bandwidth. In the experiment, a PM with a 3-dB bandwidth of 20 GHz is selected, to generate an LCMW with the highest frequency up to 18 GHz.

It is worth noting that thousands of modes oscillate at the same time. In fact, these oscillating modes have a specific phase relationship and oscillation amplitude. Consequently, a continuous microwave waveform (both in phase and amplitude) is formed.

Experimentally, the SSB phase noise of the generated microwave signal is measured by a phase noise analyzer (R&S FSWP) under single frequency oscillation condition. As shown in Fig. 7, the phase noise is as low as –134.5 dBc/Hz at 10 kHz frequency-offset. There are some spurs at the integer multiples of 45 kHz, which corresponds to the side-modes defined by the cavity round-trip time. It is noted that phase noise difference at 45 kHz and its multiples can be observed in Fig. 7. Actually, the side-modes in the cavity compete for gain, resulting in power

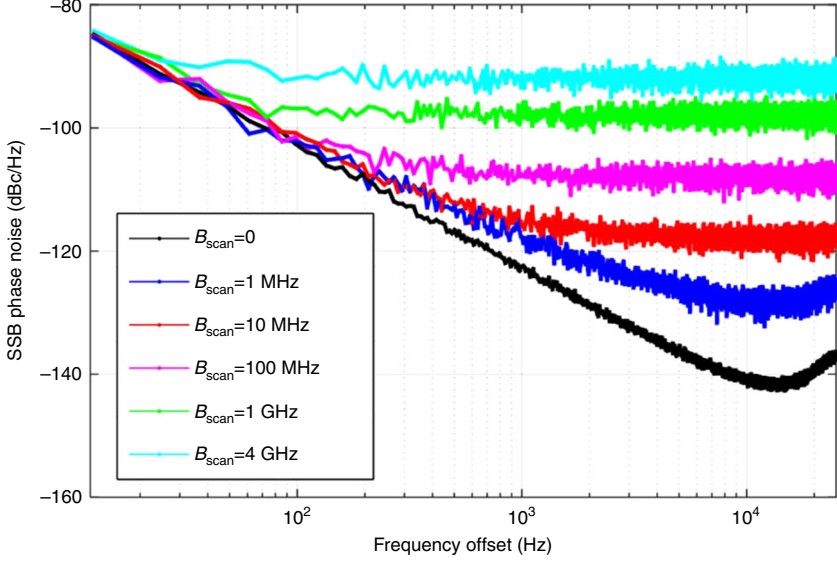

**Fig. 6** Simulated phase noise performance with different scanning bandwidth. $B_{scan} = 0$ corresponds to a conventional single mode OEO

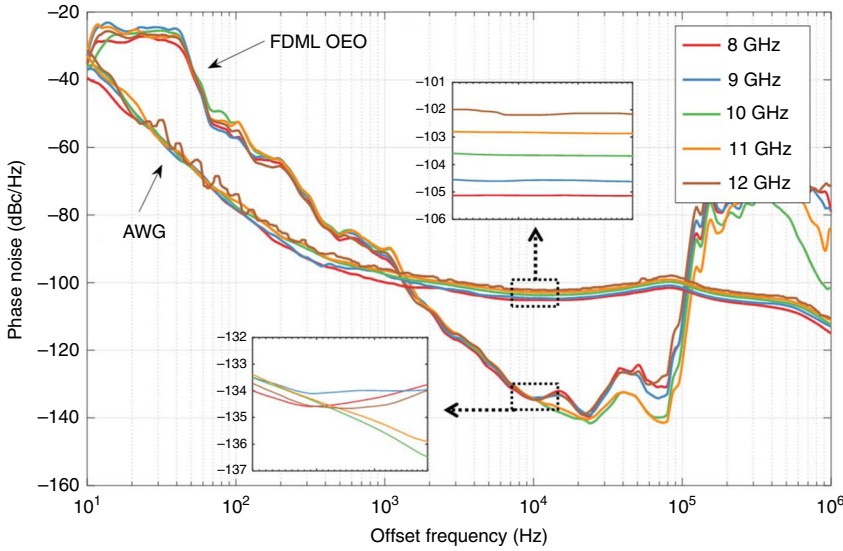

**Fig. 7** Measured phase noise performance. Phase noise of microwave signals in X-band generated by the FDML OEO and a state-of-art electronic arbitrary waveform generator (AWG) are measured. Inset shows the detailed phase noises at 10 kHz frequency offset. AWG arbitrary waveform generator

variations of the side-modes. Thus, the phase noise at around 45 kHz and its multiples changes from one measurement to another. As a comparison, the phase noise of a signal generated by a state-of-art electronic arbitrary waveform generator (Tektronix AWG70001A) is also measured and shown in Fig. 7. Significantly, the phase noise of microwave signal generated by the FDML OEO does not degrade with the increasing of microwave frequency comparing to electronics techniques[8]. The phase noise of the signal generated by the FDML OEO at 10 kHz frequency offset is lower than that generated by the AWG, but is much higher at low-frequency offset. The relative large phase noise close to the carrier is mainly caused by the ambient fluctuation because the OEO is sensitive to the environmental changes. A lower phase noise close to the carrier can be expected by using vibration and thermal isolation, as well as the phase locking technique[38]. The phase noise far from the carrier can be effectively reduced using a multi-loop OEO[39] since the side-mode spacing is increased due to the Vernier effect. See Supplementary Note 5 for further details on phase noise improvement of the

FDML OEO. In this way, the phase noise at a frequency-offset both close to and far from the carrier can be reduced.

## Discussion

In summary, we have proposed, theoretically analyzed, and experimentally demonstrated a novel OEO based on FDML to overcome the frequency tuning speed limitation of a conventional OEO. Theoretical analysis results concluded that all the modes of the FDML OEO are stably oscillating in the cavity, and are mode-locked with a fixed phase relationship among them. In addition, it can be concluded that FDML OEO oscillates in the same way as a conventional single-mode OEO which uses the same optical and electronic components, except that the energy is shared by the many oscillation modes. Experimental results demonstrated the generation of high-speed and reconfigurable frequency-swept microwave signals based on the proposed FDML OEO with a sweep speed up to 0.34 GHz/μs and a TBWP as high as 166,650. FDML OEO has great potential for microwave photonics signal generation and processing.

**Data availability**. The data that support the findings of this study are available from the corresponding author upon reasonable request.

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

## Acknowledgements

T.H. thanks N. Shi, S. Sun, X. Zhu and H. Sun for comments and discussion. This work was supported by National Natural Science Foundation of China under 61377069, 61335005, 61321063, 61090391 and 61620106013.

## Author contributions

M.L. conceived and designed the experiments, and T.H. performed the experiments. Q.C. and Y.D. conducted analytical calculations and carried out numerical simulations. T.H., M.L. and N.Z. analyzed the data. T.H., Q.C., Y.D., J.T., W.L., J.Y., M.L. and N.Z. wrote the paper. M.L. and N.Z. supervised the project.

## Additional information

**Competing interests:** The authors declare no competing interests.

