## [Peer Review File · Nature Communications]

Reviewer's Comments

Reviewer #1 (Remarks to the Author)

This paper describes a novel scheme for generation of a chirped microwave signal using the optoelectronic oscillator (OEO) architecture. The authors employ a combination of a phase modulator, an optical filter, and a tunable light source to modify the conventional OEO scheme and demonstrate a chirped x-band signal at the output of the oscillator. The bandwidth of the chirp and its repetition rate is controlled by adjusting the current of the tunable laser with a saw tooth waveform.

This is a very interesting paper with a novel scheme that puts to advantage photonic components (tunable laser, Bragg grating optical filter) and the versatility of the OEO architecture to produce a useful source for radar applications. The authors provide an analytical model and a complimentary numerical simulation to support the experimental observation. The results of the experiment is compared with an AWG.

While the experimental observations demonstrate generation of chirped microwave output with expected bandwidth and repetition rate, some of the conclusions regarding the phase noise performance of the oscillator require a more careful consideration. The authors tacitly assume that the phase noise of each line of FMDL oscillation is the same as the phase noise of a single mode OEO utilizing the same optical and electronic components in the OEO loop. This statement must be carefully considered, because in the OEO loop the power on the photodetector determines the white phase noise (up to shot noise) and the phase noise floor of the output. This power is larger in the single mode operation than in chirped operation. Indeed, this is clearly depicted in Fig. 6, where increasing the frequency of modulation results in deterioration of the phase noise floor. Furthermore, the statement that the phase noise deteriorates but SNR remains the same (line 276) is unclear: what SNR are the authors referring to, and what is its value? These are important parameters which determine the ultimate usefulness of this chirped oscillator in actual applications.

There are a few other statements that must be reconsidered. The authors refer to a nonlinearity in the produced chirp and associate this with the linearity electronic saw tooth generator. Indeed, tunable lasers, especially diode lasers, have a nonlinear response to a linear saw tooth. Virtually all electronic saw tooth generators have better linearity than the laser response. Unless care is exercised in linearization of the laser output, a better electronic source is unlikely to improve the linearity of the output. Finally, increasing the Q of the loop will only aid in improving the close to carrier phase noise of the oscillator and not its far from the carrier, as seems to be implied by the last sentence in the conclusions. This should be clarified as well.

Reviewer #2 (Remarks to the Author):

This paper proposes and experimentally demonstrates a large chirp-rate LCMW based on Fourier domain mode locking to break the limitation of mode locking time in an OEO, and achieved an LCMW with a high chirp rate of 0.34GHz/us and a large time bandwidth product of 166,650. The key novelty of this paper is to employ Fourier domain mode locking in an OEO to provide an effective solution of fast tunable microwave signals. Theoretical analysis is provided and solid experimental results are demonstrated. Generally, this is a good paper and will be interesting to the microwave photonics communities.

My main comments about this paper is as below:

1. Nature communications is aimed to cover a wide range of readers. For the researchers who are not familiar with microwave photonics or LCMW, they may not understand the importance and applications of LCMW and why the fast tuning speed and big time-bandwidth product is necessary. The authors only give a very simple sentence to address the importance of LCMW and which is far from enough.

2. There have already been a lot of publications about microwave waveform generation with large TBWP using optical methods, such as

} P. Zhou, F. Z. Zhang, Q. S. Guo and S. L. Pan, "Linearly chirped microwave waveform generation with large time-bandwidth product by optically injected semiconductor laser," *Opt. Express*, vol. 24, no. 16, pp. 18460-18467, 2016.

} J. P. Zhuang, X. Z. Li, S. S. Li and S. C. Chan, "Frequency-modulated microwave generation with feedback stabilization using an optically injected semiconductor laser," *Opt. Lett.*, vol. 41, no. 24, pp. 5764-5767, 2016

In the above *Opt. Express* 2016 paper, TBWP of 1.2×10^5 has been achieved, which is equivalent to this paper. The authors did not cite these papers and compared the benefits of the proposed technique.

3. The key novelty of this paper is to break the limitation of mode locking time in an OEO, therefore enables fast tunable speed of the generated microwave frequency. The concept of Fourier domain mode locking is borrowed from the fast tunable laser in optical domain and now is employed in electrical domain. If the authors are the first time to employ the FDML concept in electrical domain to solve a major problem in microwave photonics, it is also very valuable. I found there is a conference paper from MWP2017 which use a very similar idea to achieve the same function:

} P. Zhou, F. Z. Zhang, Q. Guo, and S.L. Pan, "Linear frequency-modulated waveform generation based on a tunable Optoelectronics Oscillator," MWP2017, Paper Mo2.1.

The concept is exactly same and the only difference is the way to generate microwave photonics filter but it is not the key of the technique. In this conference paper, the tuning speed of 0.18GHz/ns is much faster, but the TBWP is much less, which is around 2804. So there may be some tradeoff between tuning speed and TBWP. The authors are suggested to compare the results difference and explain the tradeoff between tuning speed and TBWP.

Besides, here are some specific comments and questions to help the authors to further improve the quality of this paper.

1. When using PM, how to solve the noise problem introduced by phase mismatching?
2. When directly modulating laser with current-driven sweep signal, how to solve the nonlinearity of the generated signal, which will directly affect the signal quality, and very important for LCMW. What's more, using a long loop as a delay, the loop jitter can further degrade the signal's linearity.
3. What is the tuning resolution by adjusting the DC bias to change the carrier frequency?
4. Why is there a phase noise difference at around 100 kHz offset frequency in Fig. 7, while the phase noise curves are almost coincident in other offset frequency?

Reviewer #3 (Remarks to the Author):

General

The paper „ Breaking the Limitation of Mode Building Time in an Optoelectronic Oscillator “ by Hao et al. describes an optoelectronic oscillator setup which uses the FDML concept to realize a very High-Q oscillator by the long fiber delay line. With this setup the authors demonstrate a very rapidly frequency swept waveform. The authors claim that their setup has the potential to generate much more rapidly wavelength swept microwave waveforms than conventional approaches, which are usually limited by the buildup time of oscillation. The authors also present theory and experimental characterization of the performance of the setup. Most of the paper is well written and clearly structured, however some points are very rudimentary discussed. The approach presented in this manuscript appears highly innovative and very interesting for many readers from physics to electrical engineering. It may open the path to a new generation or to a new class of microwave generators. However right now my main concern is the poor performance of the system. It looks like only over a very limited frequency range next to each comb line the phase noise is improved. I think it is mandatory to either show better data or to convincingly explain a strategy how this can be improved in the future. If this can be done, after revision, the manuscript is highly appropriate for Nature Communications.

Major points

- 1) The description of the setup and the level of detail in it is not sufficient. It would be very interesting to have all the technical data of the individual components used. Small signal gain, noise figure, amplification factors, spectral bandwidth, power levels, losses and especially in the long delay fiber the remaining dispersion values are required. I think it is mandatory to include this information to better understand the performance, the future challenges, opportunities and also the shortcomings of the design.
- 2) The characterization of the compressibility of the waveform is not clear to me. The authors calculate just an autocorrelation. However I am not sure if a simple linear autocorrelation is sufficient to determine the compressibility. It just gives information about the spectral bandwidth, doesn't it? For example in optics to determine the duration of a laser pulse it is not possible to derive it from the first order autocorrelation because a wide spectrum no matter if it consists of noise or a pulse train, both have a wideband output. Just by a nonlinear autocorrelation this can be determined. Also the section about this analysis is very very short. It would be very interesting just

to see oscilloscope traces of let's say 100 waveforms recorded in in fast frame mode of the Tektronix oscilloscope and plotted over each other. I would be interesting to see, how well they lie over each other to directly judge teim jitter. I suggest plotting a 2D statistical color-map diagram similar to the one you used in figure S3 of the supplementary material. However, not the frequency over time should be plotted but simply the electronic waveforms over time.

3) My main concern is the not too great performance of the system right now. The authors claim a much improved phase noise performance of optoelectronic oscillators, however, a superior performance is only observed in a very narrow frequency range with respect to the carrier. Only from 1kHz - ~100kHz it is better than the one from the arbitrary waveform generator. It must be discussed why this is the case and it has to be shown how to improve the performance in the future. Also, a discussion would be helpful if this performance is already usable right now. The authors also state that the performance can be improved for frequencies closer to the carrier by simply modifying the delay line. I think this is not true: If frequencies down to approximately 100 to 1000 Hz from the carrier should to be addressed (improved), it would be required to apply fiber delay lines of approximately 300-3000 km length. This is not feasible, because lost would be too high. So it is absolutely required to discuss how the performance will be improved in the future. (BTW: was the AWG set to a single frequency or did it generate a swept waveform?)

Minor points

- 1) Please give a description of all the abbreviations and labels in the figures. Many are missing.
- 2) Please include an optical spectrum of the optical branch. Ideally with very high resolution.

I suggest to invite the authors to revise their manuscript to address specific concerns before a final decision is reached.

Responses to Reviewers:

Reviewer #1

This paper describes a novel scheme for generation of a chirped microwave signal using the opto-electrooptic oscillator (OEO) architecture. The authors employ a combination of a phase modulator, an optical filter, and a tunable light source to modify the conventional OEO scheme and demonstrate a chirped x-band signal at the output of the oscillator. The bandwidth of the chirp and its repetition rate is controlled by adjusting the current of the tunable laser with a saw tooth waveform.

This is a very interesting paper with a novel scheme that puts to advantage photonic components (tunable laser, Bragg grating optical filter) and the versatility of the OEO architecture to produce a useful source for radar applications. The authors provide an analytical model and a complimentary numerical simulation to support the experimental observation. The results of the experiment is compared with an AWG.

While the experimental observations demonstrate generation of chirped microwave output with expected bandwidth and repetition rate, some of the conclusions regarding the phase noise performance of the oscillator require a more careful consideration. The authors tacitly assume that the phase noise of each line of FMDL oscillation is the same as the phase noise of a single mode OEO utilizing the same optical and electronic components in the OEO loop. This statement must be carefully considered, because in the OEO loop the power on the photodetector determines the white phase noise (up to shot noise) and the phase noise floor of the output. This power is larger in the single mode operation than in chirped operation. Indeed, this is clearly depicted in Fig. 6, where increasing the frequency of modulation results in deterioration of the phase noise floor. Furthermore, the statement that the phase noise deteriorates but SNR remains the same (line 276) is unclear: what SNR are the authors referring to, and what is its value? These are important parameters which determine the ultimate usefulness of this chirped oscillator in actual applications.

Response: Thank you for the helpful comments and suggestions. We have reconsidered the statement about the phase noise performance of the FDML OEO. The reviewer is right that the phase noise of each line of the FDML OEO is different from the phase noise of a single-mode OEO, which can be clearly observed in Fig. 6. As can be seen, at a frequency-offset that is close to the

carrier, the SSB noise of the FDML OEO is almost the same for the proposed FDML OEO different scanning bandwidths as well as for a single-mode OEO (i.e. $B_{\text{scan}}=0$), as shown in Fig. 6, because the phase noise is mainly determined by the Q-factor of the OEO cavity. At a frequency-offset far from the carrier, however, the additive noise would contribute to the most part of the phase noise. Since the total signal power is shared by the many oscillation modes, the power of one mode would decrease with the increase in the scanning bandwidth of the FDML OEO. As a result, the phase noise performance at a frequency-offset far from the carrier would get deteriorated with the increase in the scanning bandwidth.

The definition of signal-to-noise ratio (SNR) is clarified in the revised manuscript. The SNR refers to the ratio between the total signal power and the total noise power for a fixed observation bandwidth, which is 4 GHz in our manuscript. From the theoretical analysis, the SNR is the same for different scanning bandwidths of the FDML OEO. For a 4-GHz observation bandwidth, the SNR is calculated to be 73 dB for a scanning bandwidth of 4 GHz (from 0 to 4 GHz). The same SNR for different scanning bandwidths indicates that the FDML OEO oscillates in the same way as a conventional single-mode OEO, except that the energy is shared by the many oscillation modes.

We have added the above discussion to the revised manuscript in the first paragraph on page 10, copied here for convenience.

“At a frequency-offset close to the carrier, the SSB noise of the FDML OEO is identical for different scanning bandwidths as well as for a single mode OEO (i.e. $B_{\text{scan}}=0$), as shown in Fig. 6, because the phase noise is determined by the Q-factor of the OEO cavity. At a frequency-offset far from the carrier, however, additive noise would be the dominant component in the phase noise. The power of each mode would decrease with the increase in the scanning bandwidth of the FDML OEO. Thus, the phase noise performance would get deteriorated with the increase in the scanning bandwidth. The SSB noise at a frequency offset far from the carrier would increase approximately by $10\log_{10}(B_{\text{scan}}/\text{FSR})$ as compared with that of a single-mode OEO where the FSR is the free spectral range defined as the frequency interval between two adjacent modes. Although the phase noise performance at a frequency offset far from the carrier get deteriorated with the increase in the scanning bandwidth, the SNR is maintained unchanged for different scanning bandwidths. The SNR is defined as the ratio between the total signal power and the total noise power for a fixed observation bandwidth. For a 4-GHz observation bandwidth, the SNR is calculated to be 73 dB for a scanning bandwidth of 4 GHz (from 0 to 4 GHz). The same SNR for different scanning bandwidths indicates that the FDML OEO oscillates in the same way as a conventional single-mode OEO, except that the energy is shared by the many oscillation modes.”

There are a few other statements that must be reconsidered. The authors refer to a nonlinearity in the produced chirp and associate this with the linearity electronic saw tooth generator. Indeed, tunable lasers, especially diode lasers, have a nonlinear response to a linear saw tooth. Virtually all electronic saw tooth generators have better linearity than the laser response. Unless care is exercised in linearization of the laser output, a better electronic source is unlikely to improve the linearity of the output.

Response: It is true that the nonlinearity of the chirp in the generated chirped signal is mainly resulted from the nonlinear response of the laser diode to the linear saw tooth driving signal. The linearity of the chirped signal can be improved by employing a feedback controller (see Satyan, N. et al., Opt. Express. 17, 15991-15999 (2009)).

We have added the above discussion to the revised manuscript in the first paragraph on page 8.

“A slight nonlinearity of the frequency distribution of the generated chirped waveform as shown in Fig. 3d is observed, which is mainly resulted from the nonlinear response of the laser diode to the linear saw tooth-driving signal. The linearity of the chirped waveform can be improved using a feedback control loop³⁴.”

34. Satyan, N., Vasiyev, A., Rakuljic, G., Leyva, V. & Yariv, A. Precise control of broadband frequency chirps using optoelectronic feedback. Opt. Express. 17, 15991-15999 (2009).

Finally, increasing the Q of the loop will only aid in improving the close to carrier phase noise of the oscillator and not its far from the carrier, as seems to be implied by the last sentence in the conclusions. This should be clarified as well.

Response: Yes, increasing the Q factor of the OEO can only improve the phase noise performance at a frequency offset close to the carrier. We have corrected it in the revised manuscript. Moreover, we have also discussed solutions to improve the phase noise performance at a frequency offset both close to and far from the carrier.

We have added a section to discuss the reduction in phase noise in the revised manuscript, given in the last paragraph on page 11.

“The relative large phase noise close to the carrier is mainly caused by the ambient fluctuation because the OEO is sensitive to the environmental changes. A lower phase noise close to the carrier can be expected by using vibration and thermal isolation, as well as the phase locking technique³⁹. The phase noise far from the carrier can be effectively reduced using a multi-loop OEO⁴⁰ since the side-mode spacing is increased due to the Vernier effect. In this way, the phase noise at a frequency offset both close to and far from the carrier can be reduced.”

39. Zhang, L., Poddar, A. K., Rohde, U. L. & Daryoush, A. S. Comparison of optical self-phase locked loop techniques for frequency stabilization of oscillators. IEEE Photon. J. 6, 7903015 (2014).

40. Yao, X. S. & Maleki, L. Multiloop optoelectronic oscillator. IEEE J. Quantum Electron. 36, 79-84 (2000).

We have also added a section to discuss the reduction in phase noise in the revised supplementary document.

5. Phase noise improvement of the FDML-OEO

The relatively large phase noise close to the carrier is mainly caused by the ambient fluctuations because the OEO is sensitive to the environmental changes. Thus, a lower phase noise close to the carrier can be expected by using vibration and thermal isolation.

In addition, the phase locking technique, which is widely used for frequency stabilization of an oscillator, can also be used in our scheme to reduce the phase noise close to the carrier⁴. Basically, an optical self-phase locked loop (SPLL) can be used to stabilize the FDML-OEO. To do so, a portion of the OEO optical output before the PD is coupled out of the OEO loop and delayed by T_D , and the phase of the delayed signal is compared with the phase of the microwave signal generated from OEO. The delay time T_D should satisfy $T_D = lT_{round-trip}$, where $T_{round-trip}$ is the round-trip time of the OEO loop and l is an integer. In this way, an error signal is obtained without an external reference oscillator. The error signal is then fed back to the OEO loop to change the effective loop length. The phase noise performance at low offset frequencies and the long-term frequency stability

of the OEO can be improved⁴.

On the other hand, the phase noise at a frequency offset far from the carrier is affected by the side-modes of the OEO loop. The side-mode spacing is 45 kHz in our demonstrated system. The side-modes cannot be well suppressed due to the wide bandwidth of the MPF, which is normally at least tens of megahertz. A series of peaks observed from the SSB phase noise measurement, shown in Fig. 7, corresponds to the beating between two adjacent modes which is 45 kHz, and its multiples, leading to a worse phase noise performance at a frequency offset far from the carrier, as compared with the one from the arbitrary waveform generator (AWG).

A multi-loop OEO is a good candidate⁵ to obtain low phase noise at a frequency offset far from the carrier. Figure S6a shows a dual-loop OEO, with both a short loop and a long loop. The modes for the short loop, the long loop, and the dual-loop OEO are shown in Fig. S6b. The oscillation frequency should satisfy $f_{osc} = k/T_{short-loop} = m/T_{long-loop}$, where k and m are both integers and $T_{short-loop}$ and $T_{long-loop}$ are round-trip times of the short loop and long loop, respectively. It can be seen that the side-mode spacing of the OEO is increased by k and m times for a short loop and long loop OEO, respectively. So, it is expected that the phase noise performance at a frequency offset far from the carrier can be improved. In addition, in order to enable a dual-loop OEO to operate with Fourier domain mode locking (FDML), the round-trip time of the swept MPF T_{filter} should satisfy the condition such that nT_{filter} equals to the greatest common divisor of $T_{short-loop}$ and $T_{long-loop}$, where n is an integer. The sweeping period of the MPF is shortened, as compared with that of a single-loop OEO, leading to an increased chirp rate for a given scanning bandwidth.

Fig. S6. a A dual-loop OEO. **b** The modes for a short-loop, long-loop, and dual-loop OEOs. GCD: greatest common divisor.

4. Zhang, L., Poddar, A. K., Rohde, U. L. & Daryoush, A. S. Comparison of optical self-phase locked loop techniques for frequency stabilization of oscillators. IEEE Photon. J. 6, 7903015 (2014).

5. Yao, X. S. & Maleki, L. Multiloop optoelectronic oscillator. IEEE J. Quantum Electron. 36, 79-84 (2000).

I recommend the authors consider addressing these concerns before the paper is accepted for publication.

Reviewer #2

This paper proposes and experimentally demonstrates a large chirp-rate LCMW based on Fourier domain mode locking to break the limitation of mode locking time in an OEO, and achieved an LCMW with a high chirp rate of 0.34GHz/us and a large time bandwidth product of 166,650. The key novelty of this paper is to employ Fourier domain mode locking in an OEO to provide an effective solution of fast tunable microwave signals. Theoretical analysis is provided and solid experimental results are demonstrated. Generally, this is a good paper and will be interesting to the microwave photonics communities.

My main comments about this paper is as below:

1. Nature communications is aimed to cover a wide range of readers. For the researchers who are not familiar with microwave photonics or LCMW, they may not understand the importance and applications of LCMW and why the fast tuning speed and big time-bandwidth product is necessary. The authors only give a very simple sentence to address the importance of LCMW and which is far from enough.

Response: Thank you for the helpful suggestions. We have added a discussion about the importance and the potential applications of LCMWs in the Introduction section on page 2.

“A linearly chirped microwave waveform (LCMW) with a large time bandwidth product (TBWP) is widely used in modern radar and wireless communication systems¹⁻⁴. For instance, in a radar system^{1,2}, an LCMW with a large TBWP is used to increase the range resolution by performing pulse compression or matched filtering at the radar receiver, which overcomes the tradeoff between the range resolution and the detection distance existing in a traditional radar system. In addition, the use of an LCMW permits a pulse to have much longer time duration, which would provide a much higher average power, making the detection distance increased. Furthermore, the signal-to-noise ratio (SNR) can be significantly increased by matched filtering. In a frequency-hopping communication system, an LCMW with a fast tuning speed and large bandwidth is also desired to enhance the anti-reconnaissance and anti-jamming capability^{3,4}.”

2. There have already been a lot of publications about microwave waveform generation with large TBWP using optical methods, such as

□ P. Zhou, F. Z. Zhang, Q. S. Guo and S. L. Pan, "Linearly chirped microwave waveform generation with large time-bandwidth product by optically injected semiconductor laser," Opt. Express, vol. 24, no. 16, pp. 18460-18467, 2016.

□ J. P. Zhuang, X. Z. Li, S. S. Li and S. C. Chan, "Frequency-modulated microwave generation with feedback stabilization using an optically injected semiconductor laser," Opt. Lett., vol. 41, no. 24, pp. 5764–5767, 2016.

In the above Opt. Express 2016 paper, TBWP of 1.2×10^5 has been achieved, which is equivalent to this

paper. The authors did not cite these papers and compared the benefits of the proposed technique.

Response: Thank you for the helpful suggestion. The two OpEx 2016 papers have been cited as Ref. 18 and 19 in the revised manuscript. Moreover, the advantages of the proposed technique over previous works have also been addressed, given in the first paragraph on page 2.

“Recently, photonic generation of a microwave waveform with a TBWP as large as 120,000 has been demonstrated based on the period-one oscillation in an optically injected semiconductor laser¹⁸⁻²⁰. An optoelectronic feedback loop is incorporated to reduce the noise of the generated microwave waveform^{19,20}. However, the generation of period-one oscillation is much strictly restricted by the injection conditions such as frequency detuning and injection power ratio between the master and slave lasers²¹. Moreover, the frequency of the generated microwave waveform is extremely sensitive to the polarization state of the master light.”

A discussion is also added in the first paragraph on page 7.

“The chirp rate is calculated to be 0.18 GHz/ μ s, which is much slower than that reported in other literatures^{19,20}. It is noted that the chirp rate is actually tunable by changing the speed of the linear saw tooth driving signal and the fiber length of the OEO. The product between the tuning speed and TBWP equals to the square of the scanning bandwidth. For a fixed scanning bandwidth, there is a tradeoff between the tuning speed and the TBWP. A large TBWP has been chosen in our experiment since a large TBWP is always required in radar systems.”

3. The key novelty of this paper is to break the limitation of mode locking time in an OEO, therefore enables fast tunable speed of the generated microwave frequency. The concept of Fourier domain mode locking is borrowed from the fast tunable laser in optical domain and now is employed in electrical domain. If the authors are the first time to employ the FDML concept in electrical domain to solve a major problem in microwave photonics, it is also very valuable. I found there is a conference paper from MWP2017 which use a very similar idea to achieve the same function:

□ P. Zhou, F. Z. Zhang, Q. Guo, and S.L. Pan, “Linear frequency-modulated waveform generation based on a tunable Optoelectronics Oscillator,” MWP2017, Paper Mo2.1.

The concept is exactly same and the only difference is the way to generate microwave photonics filter but it is not the key of the technique. In this conference paper, the tuning speed of 0.18GHz/ns is much faster, but the TBWP is much less, which is around 2804. So there may be some tradeoff between tuning speed and TBWP. The authors are suggested to compare the results difference and explain the tradeoff between tuning speed and TBWP.

Response: The concept of our FDML OEO is different from that reported in the conference paper. Indeed, the concept of our FDML OEO was originally used in an FDML laser. The key to implement an FDML OEO/laser is to have a fast frequency scanning microwave/optical filter. In the conference paper, however, no frequency scanning microwave filter is involved. Actually, the linearly chirped microwave waveform (LCMW) in their scheme is generated by beating two optical modes from an optically injected semiconductor laser. The OEO feedback loop is only used to reduce the phase noise of the LCMW. In other words, the LCMW is not generated by the OEO. In our scheme, however, the LCMW is directly generated by the OEO. We believe this is the key difference between the two approaches.

The conference paper is cited as Ref. 20 in the revised manuscript, and a discussion is added in the revised manuscript on page 2.

“Recently, photonic generation of a microwave waveform with a TBWP as large as 120,000 has been demonstrated based on the period-one oscillation in an optically injected semiconductor laser¹⁸⁻²⁰. An optoelectronic feedback loop is incorporated to reduce the noise of the generated microwave waveform^{19,20}. However, the generation of period-one oscillation is much strictly restricted by the injection conditions such as frequency detuning and injection power ratio between the master and slave lasers²¹. Moreover, the frequency of the generated microwave waveform is extremely sensitive to the polarization state of the master light.”

The product between the tuning speed and the TBWP equals to the square of the scanning bandwidth. For a fixed scanning bandwidth, there is a tradeoff between the tuning speed and the TBWP. For example, a fast tuning speed results in a small TBWP. In our scheme, a large TBWP is realized at the expense of a slow tuning speed.

We have added a discussion to the revised manuscript in the first paragraph on page 7.

“The chirp rate is calculated to be 0.18 GHz/μs which is much slower than that reported in other literatures^{19,20}. It is noted that the chirp rate is actually tunable by changing the speed of the linear saw tooth driving signal and the fiber length of the OEO. The product between the tuning speed and the TBWP equals to the square of the scanning bandwidth. For a fixed scanning bandwidth, there is a tradeoff between the tuning speed and the TBWP. A large TBWP has been chosen in our experiment since an LCMW with a large TBWP is always required in radar systems.”

Besides, here are some specific comments and questions to help the authors to further improve the quality of this paper.

1. When using PM, how to solve the noise problem introduced by phase mismatching?

Response: Phase matching between the microwave and optical signals is one of the basic requirements for broadband and low noise operation of a phase modulator (PM). Consider a single frequency electrical drive signal

$$V(z, t) = V_0 \sin 2\pi f \left(\frac{zn_m}{c} - t \right)$$

where f is the microwave frequency, n_m is the refractive index, and c is the speed of light in vacuum. The phase shift $\Delta\varphi(f, t)$ of the optical wave at the output of the modulator is given by

$$\Delta\varphi(f, t) = \Delta\beta_0 L \frac{\sin\left(\frac{\pi f n_m \delta L}{c}\right)}{\frac{\pi f n_m \delta L}{c}} \sin 2\pi f \left(\frac{n_m \delta L}{2c} - t \right)$$

where $\delta = 1 - n_0/n_m$ represents the refractive index difference between the microwave and optical signals, n_0 is the refractive index of the guided optical mode, $\Delta\beta_0$ is the amplitude change in wave vector, and L is the interaction length. It can be seen that a phase attenuation factor

$\frac{\sin\left(\frac{\pi f n_m \delta L}{c}\right)}{\frac{\pi f n_m \delta L}{c}}$ as a function of frequency f is introduced. For $n_0 \neq n_m$, the optical and microwave waves are not velocity matched, which makes the modulation efficiency low and highly dependent on the microwave frequency, leading to the PM to have a small 3-dB modulation bandwidth.

The PM used in our experiment is a titanium-diffused lithium niobate 20 GHz commercially available device (Photline, MPZ-LN-20). The phase matching is acceptable within the modulation bandwidth of the PM. However, the slight phase mismatch of the PM and the uneven magnitude response of the electrical amplifier and the photodetector (PD) would lead to the ripples of the

electrical spectrum of the generated microwave signal. As discussed in the theoretical analysis, the phase noise of the generated microwave signal at a frequency offset close to the carrier is determined by the Q factor of the OEO, which is generally independent of the phase mismatch of the PM. However, the additive noise would contribute to the most of the phase noise at a frequency offset far from the carrier. An OEO loop with a smaller bandwidth would result in a higher phase noise at a frequency-offset far from the carrier. Thus, the phase noise at a frequency-offset far from the carrier is affected by the uneven magnitude response of the OEO loop including the phase mismatching of the PM.

In order to reduce the impact of the phase mismatching on the phase noise of the generated microwave signal, we chose a PM with a sufficiently wide bandwidth (a 3-dB bandwidth of 20 GHz) to generate a chirped microwave signal with a highest frequency up to 18 GHz.

We have added a discussion to address this issue in the revised manuscript, in the first paragraph on page 10.

“The phase mismatch between the optical and microwave signals in the PM would result in an uneven power distribution of the generated microwave waveform, leading to an increased phase noise at a high frequency offset. In order to reduce the impact of the phase mismatch on the phase noise of the generated microwave signal, we chose a PM with a sufficiently wide bandwidth. In the experiment, a PM with a 3-dB bandwidth of 20 GHz is selected, to generate an LCMW with the highest frequency up to 18 GHz.”

2. When directly modulating laser with current-driven sweep signal, how to solve the nonlinearity of the generated signal, which will directly affect the signal quality, and very important for LCMW. What's more, using a long loop as a delay, the loop jitter can further degrade the signal's linearity.

Response: The nonlinearity of the chirp in the generated chirped signal is mainly resulted from the nonlinear response of the laser diode to the linear saw tooth driving signal. The linearity of the chirped signal can be improved by employing a feedback controller (see Satyan, N. et al., Opt. Express. 17, 15991-15999 (2009)).

We have added the above discussion to the revised manuscript, in the first paragraph on page 8.

“A slight nonlinearity of the frequency distribution of the generated chirped waveform as shown in Fig. 3d is observed, which is mainly resulted from the nonlinear response of the laser diode to the linear saw tooth-driving signal. The linearity of the chirped waveform can be improved using a feedback control loop³⁴.”

34. Satyan, N., Vasiyev, A., Rakuljic, G., Leyva, V. & Yariv, A. Precise control of broadband frequency chirps using optoelectronic feedback. Opt. Express. 17, 15991-15999 (2009).

3. What is the tuning resolution by adjusting the DC bias to change the carrier frequency?

Response: In principle, the carrier frequency can be continuously tuned by adjusting the DC bias. In our experiment, however, the tuning resolution of the carrier frequency is around 5 MHz, limiting by the tuning resolution of the DC source. We have added this point to the revised manuscript, in the second paragraph on page 8.

“The frequency tuning resolution is around 5 MHz in our experiment, which is limited by the

resolution of the DC driving source.”

4. Why is there a phase noise difference at around 100 kHz offset frequency in Fig. 7, while the phase noise curves are almost coincident in other offset frequency?

Response: The side-mode spacing of our OEO loop is 45 kHz. So, a series of phase noise peaks can be observed at 45 kHz and its multiples. Actually, not only phase difference at around 100 kHz frequency offset can be seen, but also at 45 kHz and its multiples. The side-modes in the cavity compete for gain, resulting in power variations of the side-modes. Thus, phase noise at around 45 kHz and its multiples changes from one measurement to another. We believe this is the reason why there is phase noise difference at 45 kHz and its multiples (including around 100 kHz). The phase noise at around 45 kHz and its multiples can be reduced by a multi-loop OEO, which has been discussed in response to Reviewer #3, Question#3.

We have added the above discussions in the revised manuscript, in the second paragraph on page 11.

“It is noted that phase noise difference at 45 kHz and its multiples can be observed in Fig. 7. Actually, the side-modes in the cavity compete for gain, resulting in power variations of the side-modes. Thus, the phase noise at around 45 kHz and its multiples changes from one measurement to another.”

Reviewer #3

General

The paper “Breaking the Limitation of Mode Building Time in an Optoelectronic Oscillator” by Hao et al. describes an optoelectronic oscillator setup which uses the FDML concept to realize a very High-Q oscillator by the long fiber delay line. With this setup the authors demonstrate a very rapidly frequency swept waveform. The authors claim that their setup has the potential to generate much more rapidly wavelength swept microwave waveforms than conventional approaches, which are usually limited by the buildup time of oscillation. The authors also present theory and experimental characterization of the performance of the setup. Most of the paper is well written and clearly structured, however some points are very rudimentary discussed. The approach presented in this manuscript appears highly innovative and very interesting for many readers from physics to electrical engineering. It may open the path to a new generation or to a new class of microwave generators. However right now my main concern is the poor performance of the system. It looks like only over a very limited frequency range next to each comb line the phase noise is improved. I think it is mandatory to either show better data or to convincingly explain a strategy how this can be improved in the future. If this can be done, after revision, the manuscript is highly appropriate for Nature Communications.

Major points

1) The description of the setup and the level of detail in it is not sufficient. It would be very interesting to have all the technical data of the individual components used. Small signal gain, noise figure, amplification factors, spectral bandwidth, power levels, losses and especially in the long delay fiber the remaining dispersion values are required. I think it is mandatory to include this information to better understand the performance, the future challenges, opportunities and also the shortcomings of the design.

Response: Thank you for the helpful suggestions. The optical carrier is provided by a distributed

feed-back (DFB) laser diode with 13 dBm output power. A laser diode controller (ILX 3724C) and microwave signal generator (Agilent 81150A) are used to drive the laser diode. The bandwidth of the phase modulator (PM) (Photline) is 20 GHz. An erbium-doped fiber amplifier (EDFA, JDS Uniphase) with a small signal gain of 37 dB and a noise figure of 3.3 dB is used to amplify the signal at the output of the notch filter. The optical fiber has a residual dispersion of -0.848 ps/nm and total insertion loss of 3.8 dB. The photodetector (Agilent) has a 3-dB bandwidth of 15 GHz and a conversion gain of 300 V/W. An electrical amplifier with a gain of 22 dB and a noise figure of 6 dB is used to boost the power of the microwave signal.

We have a discussion in which all these parameters are provided in the revised manuscript, in the first paragraph on page 4.

“The long-time delay line is a dispersion-free 4.5 km optical fiber consisting of a single-mode fiber and a dispersion compensating fiber with opposite dispersion of the same magnitude. The optical fiber has a residual dispersion of -0.848 ps/nm and total insertion loss of 3.8 dB. To implement the MPF, a light wave from a tunable laser source (TLS) with 13 dBm output power is sent to a 20-GHz phase modulator (PM). A phase-shifted fiber Bragg grating (PS-FBG) with an ultra-narrow notch is used to remove one of the two sidebands. The reflection spectrum of the PS-FBG is given in Fig. 2b, where the inset shows the magnitude and phase responses of the PS-FBG around the notch. The notch is located at around 1550.6 nm with a single phase cycle as shown in the phase response. The output light wave from the PS-FBG is amplified by an erbium-doped fiber amplifier (EDFA, JDS Uniphase) with a small signal gain of 37 dB and a noise figure of 3.3 dB. A PD with a 3-dB bandwidth of 15 GHz and a conversion gain of 300 V/W is used to convert the optical signal to an electrical signal. The overall operation is equivalent to a high-Q MPF, implemented based on phase modulation, and phase-modulation to intensity-modulation (PM-IM) conversion^{31, 32}. The frequency tuning of the MPF is achieved by sweeping the wavelength of the TLS which is driven by a sawtooth current. An electrical amplifier with a gain of 22 dB and a noise figure of 6 dB is used to boost the power of the electrical signal. The amplified electrical signal is fed back to the PM to form a closed loop.”

2) The characterization of the compressibility of the waveform is not clear to me. The authors calculate just an autocorrelation. However I am not sure if a simple linear autocorrelation is sufficient to determine the compressibility. It just gives information about the spectral bandwidth, doesn't it? For example in optics to determine the duration of a laser pulse it is not possible to derive it from the first order autocorrelation because a wide spectrum no matter if it consists of noise or a pulse train, both have a wideband output. Just by a nonlinear autocorrelation this can be determined. Also the section about this analysis is very very short. It would be very interesting just to see oscilloscope traces of let's say 100 waveforms recorded in in fast frame mode of the Tektronix oscilloscope and plotted over each other. I would be interesting to see, how well they lie over each other to directly judge them jitter. I suggest plotting a 2D statistical color-map diagram similar to the one you used in figure S3 of the supplementary material. However, not the frequency over time should be plotted but simply the electronic waveforms over time.

Response: Yes, autocorrelation is generally used in optics to measure the duration of an ultrashort laser pulse that is too short to measure directly. However, the time duration of our chirped microwave waveform is long enough to be directly observed using an oscilloscope. The reason that we calculated the autocorrelation of the chirped microwave waveform is to show the pulse compression. The temporal compression process is usually performed using a matched filter with an impulse-response equals to the complex conjugate of the time reversal of the received signal, thus the output of the matched filter is identical to the autocorrelation of the chirped microwave waveform.

We have added a discussion to the revised manuscript in the first paragraph on page 8.

“In modern radars, pulse compression is generally used to increase the range resolution. This is normally realized using a matched filter with an impulse-response corresponding to the complex conjugate of the time reversal of the received signal. The output of the matched filter is identical to the autocorrelation of the chirped microwave waveform. In order to see the pulse compression capability, we calculated the autocorrelation of the generated LCMW, and the result is shown in Fig. 3e. The width of the compressed pulse is around 0.275 ns, corresponding to a pulse compression ratio of 80800.”

The oscilloscope traces of 12 waveforms recorded in the fast frame mode of the Tektronix oscilloscope is shown below. The span was set to be 40 μs in order to cover at least one period of the generated waveform. The sampling rate was set to be 25 GS/s rather than the maximum 100GS/s in order to have more frames overlay together due to the limited memory depth of the oscilloscope. Unfortunately, only 12 traces were recorded even at a low sampling rate. Only small jitters can be seen from Fig. S4, which indicates good consistency of the generated microwave waveforms. We have added the oscilloscope traces recorded in fast frame mode and discussions about it in the revised supplementary document.

Fig. S4. Overlay of 12 traces for frequency scanning microwave waveforms from 8 to 12 GHz recorded in the Fast Frame mode of the Tektronix oscilloscope. a In a span of 40 μs . b 100,000 times zoom-in view.

3. Signal Consistency of the of the FDML-OEO

An overlay of 12 traces for frequency scanning microwave waveforms from 8 to 12 GHz recorded in the Fast Frame mode of the Tektronix oscilloscope are shown in Fig. S4a. The span was set to be 40 μs in order to trace at least one period of the generated waveform. The sampling rate was set to be 25 GS/s rather than 100 GS/s in order to have more frames overlay. The 100,000 times zoom-in

view of the overlaid 12 traces is shown in Fig. S4b. Only very small jitters can be seen from Fig. S4, which indicates a good consistency of the generated chirped microwave waveform.

3) My main concern is the not too great performance of the system right now. The authors claim a much improved phase noise performance of optoelectronic oscillators, however, a superior performance is only observed in a very narrow frequency range with respect to the carrier. Only from 1kHz - ~100kHz it is better than the one from the arbitrary waveform generator. It must be discussed why this is the case and it has to be shown how to improve the performance in the future. Also, a discussion would be helpful if this performance is already usable right now. The authors also state that the performance can be improved for frequencies closer to the carrier by simply modifying the delay line. I think this is not true: If frequencies down to approximately 100 to 1000 Hz from the carrier should to be addressed (improved), it would be required to apply fiber delay lines of approximately 300-3000 km length. This is not feasible, because loss would be too high. So it is absolutely required to discuss how the performance will be improved in the future. (BTW: was the AWG set to a single frequency or did it generate a swept waveform?)

Response: Thank you for the helpful comments and suggestions. The relatively large phase noise close to the carrier is mainly caused by the ambient fluctuation because the OEO is sensitive to the environmental changes. Thus, a lower phase noise close to the carrier can be expected by using vibration and thermal isolation.

In addition, the phase locking technique, which is widely used for frequency stabilization of an oscillators, can also be used in our scheme to reduce the phase noise close to the carrier (see e.g. Zhang, L. et al., IEEE Photon. J. 6, 7903015 (2014)). Basically, an optical self-phase locked loop (SPLL) can be used to stabilize the FDML-OEO. To do so, a portion of the OEO optical output before the PD is coupled out of the OEO loop and delayed by T_D , and the phase of the delayed signal is compared with the phase of the microwave signal generated from OEO. The delay time T_D should satisfy $T_D = lT_{round-trip}$, where $T_{round-trip}$ is the round-trip time of the OEO loop and l is an integer. In this way, an error signal is obtained without an external reference oscillator. The error signal is then feed back to the OEO loop to change the effective loop length. The phase noise performance at low offset frequencies and the long-term frequency stability of the OEO can be improved.

On the other hand, the phase noise far from the carrier is affected by the side-modes of the OEO loop. The frequency interval between the adjacent side-modes is 45 kHz in our demonstrated system. The side-modes cannot be well suppressed due to the wide bandwidth of the microwave photonic filter (MPF), which is normally at least tens of megahertz. A series of phase noise peaks observed from the SSB phase noise measurement, shown in Fig. 7, corresponds to the beating between two adjacent modes which is 45 kHz, and its multiples, leading to a worse phase noise performance at a frequency-offset far from the carrier, as compared with the one from the AWG.

A multi-loop OEO is a good candidate to obtain low phase noise at a frequency-offset far from the carrier. Figure S6a shows a dual-loop OEO, with both a short loop and a long loop. The modes for the short loop, the long loop, and the dual-loop OEO are shown in Fig. S6b. The oscillation modes for the short loop, the long loop, and the dual-loop OEO are shown in Fig. S6b. The oscillation frequency should satisfy $f_{osc} = k/T_{short-loop} = m/T_{long-loop}$, where k and m are both integers and $T_{short-loop}$ and $T_{long-loop}$ are round-trip time of the short loop and long loop, respectively. It is can be seen that the side-mode spacing of the OEO is increases by k and m times for a short-loop and long-loop OEO, respectively. So, it is expected that the phase noise performance at a frequency-offset far from the carrier can be improved. In addition, in order to enable a dual-loop OEO to operate with Fourier domain mode locking (FDML), the round-trip time of the swept MPF T_{filter} should satisfy the

condition such that nT_{filter} equals to the greatest common divisor of $T_{\text{short-loop}}$ and $T_{\text{long-loop}}$, where n is an integer. The sweeping period of the MPF is shortened, as compared with that of a single-loop OEO, leading to an increased chirp-rate for a given scanning bandwidth. The tradeoff between the chirp-rate and the TBWP has been discussed in response to Reviewer#2, Question#3.

The AWG was set to single frequency in order to evaluate the phase noise performance.

We have added the above discussions to the revised manuscript in the last paragraph on page 11.

“The relative large phase noise close to the carrier is mainly caused by the ambient fluctuation because the OEO is sensitive to the environmental changes. A lower phase noise close to the carrier can be expected by using vibration and thermal isolation, as well as the phase locking technique³⁹. The phase noise far from the carrier can be effectively reduced using a multi-loop OEO⁴⁰ since the side-mode spacing is increased due to the Vernier effect. In this way, the phase noise at a frequency offset both close to and far from the carrier can be reduced.”

39. Zhang, L., Poddar, A. K., Rohde, U. L. & Daryoush, A. S. Comparison of optical self-phase locked loop techniques for frequency stabilization of oscillators. IEEE Photon. J. 6, 7903015 (2014).

40. Yao, X. S. & Maleki, L. Multiloop optoelectronic oscillator. IEEE J. Quantum Electron. 36, 79-84 (2000).

We have also added a section to discuss the reduction in phase noise in the revised supplementary document.

5. Phase noise improvement of the FDML-OEO

The relatively large phase noise close to the carrier is mainly caused by the ambient fluctuations because the OEO is sensitive to the environmental changes. Thus, a lower phase noise close to the carrier can be expected by using vibration and thermal isolation.

In addition, the phase locking technique, which is widely used for frequency stabilization of an oscillator, can also be used in our scheme to reduce the phase noise close to the carrier⁴. Basically, an optical self-phase locked loop (SPLL) can be used to stabilize the FDML-OEO. To do so, a portion of the OEO optical output before the PD is coupled out of the OEO loop and delayed by T_D , and the phase of the delayed signal is compared with the phase of the microwave signal generated from OEO. The delay time T_D should satisfy $T_D = lT_{\text{round-trip}}$, where $T_{\text{round-trip}}$ is the round-trip time of the OEO loop and l is an integer. In this way, an error signal is obtained without an external reference oscillator. The error signal is then fed back to the OEO loop to change the effective loop length. The phase noise performance at low offset frequencies and the long-term frequency stability of the OEO can be improved⁴.

On the other hand, the phase noise at a frequency offset far from the carrier is affected by the side-modes of the OEO loop. The side-mode spacing is 45 kHz in our demonstrated system. The side-modes cannot be well suppressed due to the wide bandwidth of the MPF, which is normally at least tens of megahertz. A series of peaks observed from the SSB phase noise measurement, shown in Fig. 7, corresponds to the beating between two adjacent modes which is 45 kHz, and its multiples, leading to a worse phase noise performance at a frequency offset far from the carrier, as compared with the one from the arbitrary waveform generator (AWG).

A multi-loop OEO is a good candidate⁵ to obtain low phase noise at a frequency offset far from the

carrier. Figure S6a shows a dual-loop OEO, with both a short loop and a long loop. The modes for the short loop, the long loop, and the dual-loop OEO are shown in Fig. S6b. The oscillation frequency should satisfy $f_{osc} = k/T_{short-loop} = m/T_{long-loop}$, where k and m are both integers and $T_{short-loop}$ and $T_{long-loop}$ are round-trip times of the short loop and long loop, respectively. It can be seen that the side-mode spacing of the OEO is increased by k and m times for a short loop and long loop OEO, respectively. So, it is expected that the phase noise performance at a frequency offset far from the carrier can be improved. In addition, in order to enable a dual-loop OEO to operate with Fourier domain mode locking (FDML), the round-trip time of the swept MPF T_{filter} should satisfy the condition such that nT_{filter} equals to the greatest common divisor of $T_{short-loop}$ and $T_{long-loop}$, where n is an integer. The sweeping period of the MPF is shortened, as compared with that of a single-loop OEO, leading to an increased chirp rate for a given scanning bandwidth.

Fig. S6. a A dual-loop OEO. **b** The modes for a short-loop, long-loop, and dual-loop OEOs. GCD: greatest common divisor.

4. Zhang, L., Poddar, A. K., Rohde, U. L. & Daryoush, A. S. Comparison of optical self-phase locked loop techniques for frequency stabilization of oscillators. *IEEE Photon. J.* 6, 7903015 (2014).

5. Yao, X. S. & Maleki, L. Multiloop optoelectronic oscillator. *IEEE J. Quantum Electron.* 36, 79-84 (2000).

Minor points

1) Please give a description of all the abbreviations and labels in the figures. Many are missing.

Response: We have added the descriptions of the abbreviations and labels in the figures in the

manuscript and supplementary document.

The revision in the manuscript is copied for your convenience.

Fig. 1 Schematic to show the operations of a conventional *optoelectronic oscillator (OEO)* and an OEO based on *Fourier domain mode locking (FDML)*. a A conventional single-frequency OEO, only one mode is active in the cavity. b An OEO based on FDML for generation of a microwave signal with fast frequency tuning, all modes are active in the cavity. *E/O: electrical to optical conversion; O/E: optical to electrical conversion.*

Fig. 2 Experimental setup and microwave photonic band-pass filtering principle. b Reflection spectrum of the *phase-shifted fiber Bragg grating (PS-FBG)*

Figure 3. Experimental results. a, Spectrum of a generated X-band frequency-scanning microwave waveform with a span of 10 GHz; b, Spectrum with a span of 200 kHz; c, Temporal waveform of the periodically and continuously chirped microwave waveform, the inset shows a section of the waveform; d, Real-time frequency distribution; e, the compressed pulse by autocorrelation (inset: zoom-in display). Some missing labels has been added.

Fig. 6 A comparison of SSB noise between a conventional OEO and an OEO based on FDML. SSB: single-sideband. Some missing labels has been added.

Fig. 7 Phase noise of microwave signals in X-band generated by the FDML OEO and a state-of-art electronic AWG, inset shows the detailed phase noises at 10 KHz frequency offset. AWG: arbitrary waveform generator. Some missing labels has been added.

The changes made in the revised supplementary document are copied here for your convenience:

Fig. S1. Optical spectrum evolution when phase-modulated light wave passes through notch filter. a Traditionally explanation. b Equivalent model.

2) Please include an optical spectrum of the optical branch. Ideally with very high resolution.

Response: Thanks for the reviewer’s helpful suggestions. We have included optical spectra from the optical branch after the phase-shifted fiber Bragg grating (PS-FBG) notch filter for OEO operating at 10 GHz single-mode, 4-6 GHz linearly chirped microwave waveform (LCMW), 8-12 GHz LCMW, and 12-14 GHz LCMW cases with a resolution of 0.02 nm. Please see revised supplementary document.

4. Optical Spectrums of the FDML-OEO

The optical spectrums after the phase-shifted fiber Bragg grating (PS-FBG) notch filter are shown in Fig. S5 for the OEO operating at 10 GHz single-mode, 4-6 GHz linearly chirped microwave waveform (LCMW), 8-12 GHz LCMW, and 12-14 GHz LCMW cases, with a resolution of 0.02 nm for all cases. The first-order sideband at the low frequency side was suppressed by the PS-FBG, leaving the first-order sideband at the high frequency side and the optical carrier. The beating between the first-order sideband and the optical carrier generates a microwave signal.

Fig. S5. Optical spectra from the optical branch after the phase-shifted fiber Bragg grating (PS-FBG) notch filter when the OEO operating at different cases. a 10 GHz single-mode. b 4-6 GHz linearly chirped microwave waveform (LCMW). c 8-12 GHz LCMW. d 2-14 GHz LCMW.

I suggest to invite the authors to revise their manuscript to address specific concerns before a final decision is reached.

Reviewer's Comments

Reviewer #1 (Remarks to the Author):

I have carefully reviewed the response of the authors to my comments, and the comments of other reviewers. The modifications made to the manuscript have made the paper much stronger, and have improved its technical content. I recommend the publication of the modified manuscript and feel confident that it will be of interest to the research community.

Reviewer #3 (Remarks to the Author):

In the revised version of the paper „Breaking the Limitation of Mode Building Time in an Optoelectronic Oscillator“ by Hao et al. the authors have added substantial additional information. In my view, all the concerns of myself and the other reviewers have been addressed appropriately. The supplementary material contains additional data, and the manuscript discusses the issues and the limitations of the proposed concept in more detail. I recommend publication. I believe that the paper now can be interesting for a somewhat wider audience, and spur additional research into this new direction.

Reviewer #1

I have carefully reviewed the response of the authors to my comments, and the comments of other reviewers. The modifications made to the manuscript have made the paper much stronger, and have improved its technical content. I recommend the publication of the modified manuscript and feel confident that it will be of interest to the research community.

Response: Thank you for the positive comments and recommendation. We would like to take this opportunity to thank you again for the efforts and valuable suggestions.

Reviewer #3

In the revised version of the paper „Breaking the Limitation of Mode Building Time in an Optoelectronic Oscillator“ by Hao et al. the authors have added substantial additional information. In my view, all the concerns of myself and the other reviewers have been addressed appropriately. The supplementary material contains additional data, and the manuscript discusses the issues and the limitations of the proposed concept in more detail. I recommend publication. I believe that the paper now can be interesting for a somewhat wider audience, and spur additional research into this new direction.

Response: Thank you for the helpful comments and recommendation. We would like to take this opportunity to thank you again for the efforts and valuable suggestions.